# The Race for COVID-19 Vaccines: Accelerating Innovation, Fair Allocation and Distribution

**DOI:** 10.3390/vaccines10091450

**Published:** 2022-09-02

**Authors:** Rutger Daems, Edith Maes

**Affiliations:** Planet Strategy Group, L-9990 Luxembourg, Luxembourg

**Keywords:** COVID-19, pandemics, vaccines, innovation, procurement, allocation, patents, governance, ethics

## Abstract

Racing to develop and distribute a vaccine against COVID-19 has proven to be a challenging endeavor. Not only has there been the enormous scientific-technical challenge of developing the world’s first vaccines against a coronavirus, the subsequent ethical issues involved in vaccine allocation have been equally complex. This contribution focuses on the policy and strategic aspects of accelerating innovation, fair allocation, and distribution. We take a holistic approach to describing the various tasks and how they are interconnected. Through comparative analysis we explore the issues through the lens of multiple stakeholders: i.e., public health authorities, governments of industrialized and developing countries, and industry. At each step of the consecutive process—from vaccine development to delivery—common ground has to be found between global community stakeholders, to move the project to the next stage, until completion: (1) accelerated innovation, (2) centralized purchasing, (3) selecting suppliers, (4) equitable allocation, (5) global access and affordability, (6) compulsory vs. voluntary licensing, and (7) a universal pandemic treaty. Conclusions: Public–private partnership is essential with regards to inventing and allocating new vaccines to fight infectious disease pandemics. The nature of pandemics requires collaboration at both the national and international levels. Seven important lessons have been identified that we can learn from based on the experience gathered during the COVID-19 pandemic. We trust that these findings will be helpful in drafting preparedness guidelines and a global pandemic treaty to manage future pandemics.

## 1. Introduction

The rapid development of the pandemic caused by the SARS-CoV-2 virus has taken the world by surprise and caused enormous social and economic havoc. The world was not prepared for this, in spite of promises made ten years ago after the influenza pandemics. Is history repeating itself and are we making the same mistakes as in the past? Vaccines are crucial in this fight. However, did we invest enough in stepping up innovation, and can health systems around the globe cope with this threat?

Two years into the pandemic, it is time to investigate whether the vaccines against this new virus were developed and deployed successfully and what the advantages and drawbacks were in the policy choices made along the trajectory of vaccine development and delivery. Where other studies may focus on the innovation aspects, or the intricacies of health systems, we take a holistic view, so as to examine the subject across the whole trajectory. We are particularly interested in the interaction between the components and the relation between upstream and downstream strategic decisions. Even if new vaccines are developed, and despite their complexity, they are only useful if people worldwide get vaccinated. The extraordinary threat of the pandemic requires public–private partnership in an optimal way.

## 2. Materials and Methods

This is an applied policy research article. It offers an inter-disciplinary framework for pandemic-related vaccine development, deployment, and governance. We used a descriptive methodology to identify a number of key issues that have been encountered across the time-based value chain, from innovation to national and international allocation and delivery.

Through a subsequent series of comparative analyses aimed at highlighting the viewpoints of public and private stakeholders, these issues are further explored. In addition, possible solutions are proposed, to foster collaboration between key actors: i.e., public health authorities, governments of industrialized and developing countries, and the vaccine industry.

The critical issues are grouped into seven chapters: 1. accelerating vaccine development; 2. centralized procurement and alliances; 3. supply tendering, quality, and price; 4. global accessibility and affordability; 5. equitable allocation and distribution; 6. compulsory licensing; and 7. a universal pandemic treaty. At the end of each chapter, lessons learned are derived for that topic. The closing chapter summarizes policy and strategic recommendations.

## 3. Results

### 3.1. Accelerating Vaccine Development

#### 3.1.1. R&D Pipeline

The development, testing, and approval of several vaccines against SARS-CoV-2 in less than one year has been one of the great achievements in society’s battle against COVID-19. In retrospect, mRNA-based vaccines, in particular, were developed rapidly and have been accepted by the EMA and FDA as safe and effective, although initially not against all mutations of the virus. Being synthetic, their production technology has proven more robust than classic biological vaccine production technologies [1]. This achievement is extraordinary, because mRNA as a novel technology for the production of vaccines did not exist prior to the pandemic. The incumbent technologies relied on inactivated and live attenuated viral technologies, which were inconceivable as technologies against a coronavirus. Public research into mRNA feasibility had been underfunded for years [2]. Despite this gap, researchers at smaller biotech companies such as Moderna, BioNTech, and CureVac had been actively pursuing research in this area for more than a decade. The pandemic suddenly put mRNA development and clinical testing in a high gear, as governments started to reprioritize subsidies and partnership. Furthermore, the collaboration between pharmaceutical multinationals and biotech firms accelerated the process and brought expertise to the partnership, in scaling activities and a wide-reaching distribution network.

#### 3.1.2. Push and Pull

In support of innovative vaccine development, push and pull incentive mechanisms were designed and rapidly implemented. These mechanisms involve public–private partnership in research and development, constructing production facilities, conducting multiple clinical trials in several geographic areas and implementing tax breaks (all examples of “push” funding), as well as guaranteed new vaccine purchase commitments underwritten by governments (so called “pull” funding, as it comes at the end of the new product development chain). For details about these “advance purchase agreements”—see Section 3.2.3.

On 16 March 2020, the G7 committed to supporting the launch of joint research projects for treatments and vaccines against SARS-CoV-2 [3]. The United States (US), United Kingdom (UK), and European Union (EU) were among the first movers investing in mRNA vaccine research. Subsequently, a real breakthrough in public policy occurred when the US launched its “Operation Warp Speed” with direct support from the White House and under the auspices of BARDA (Biomedical Advanced Research and Development Authority) [4]. The UK quickly followed suit, and after coming under criticism for falling short in matching the US and UK’s support for vaccines and treatments against SARS-CoV-2, the European Commission on 16 September 2020, announced it was tabling a legislative proposal to establish a European biomedical R&D agency modeled after the US’s BARDA [5].

#### 3.1.3. Risk Mitigation

From a capital investment standpoint and in analogy with classic financial portfolio theory, government sponsors can reduce their exposure to “unsystematic risk” by simultaneously investing in various vaccine asset classes (mRNA, adenovirus vector platforms, recombinant adjuvanted technology, etc.). While diversification among classes is no guarantee against potential losses, investing in a broad number of “existing classes” and in “new science” technology is a prudent strategy, which in the case of brand-new science classes, such as mRNA, turned out to be successful. R&D-based firms are required to invest and compete against each of these asset classes and newcomers will compete with incumbent firms.

Companies try to decide early on which specific science-based platform to invest in to gain a competitive advantage. However, given the constraints on resources and governance capacity, many contenders—even the larger multinationals—are not able to invest scarce corporate resources into a great variety of risky projects. Consequently, this means that unsystematic risk (i.e., company-specific risk) cannot be mitigated by a single firm, by applying diversification across a variety of development pathways. Project returns on investment depend on achieving technical success and launching a vaccine that generates revenues.

In summary, the front-loaded funding provided by governments (push subsidies) was shown to substantially accelerate the new vaccine development process. However, R&D-based companies still assume the risk associated with technical success or failure. The advantage of such government funding is that it compensates for the “crunch time” in launching innovations earlier than otherwise would be possible. Accelerating the project development time demands extra resources with regard to skilled labor, infrastructure, and “at risk” financing. At present, frontloaded funding does not include ethical and contractual requirements regarding prioritization and allocation when a vaccine is successfully developed and launched.

#### 3.1.4. Truncated Pathway

The governance of new vaccine development can be seen as a series of “stage-gate” managerial decisions to be taken in the course of the project’s trajectory. Historically, the chance of successfully developing a new biomedical product is small and the investment is high [6].

A shorter development pathway requires a number of phases of the product development process to be conducted in parallel, instead of sequentially. The inherent risk is that such a high pressure approach significantly increases the chances of project failure along the trajectory. Above and beyond the enhanced risk of technical failure, the financial risk increases exponentially, as the firm is deprived of managerial options to mitigate the investment risk.

Classic investment theory and practice that uses “real options” allows for executive decisions to either expand, defer, or wait; or to abandon the development project altogether. A vaccine company that faces obstacles, which at a certain moment are deemed unsurmountable, can use these flexibilities to diminish the investment risk [7]. However, such risk hedging is no longer available under an accelerated timeframe and this requires making a trade-off. If the project is successful, a vaccine can be launched early. Meanwhile, if intermediate steps fail, one has to go back several steps, at the expense of time. In the worst case scenario, a vaccine development project may fail entirely, as indeed happened during the COVID-19 pandemic.

#### 3.1.5. Key Learnings

An exceptional innovation speed has marked the development of vaccines against SARS-CoV-2. The mRNA experience suggests that a vibrant vaccine ecosystem cannot be taken for granted in terms of delivering at high speed the breakthrough innovations that are needed to stop pandemics. Government intervention and support that uses push and pull incentive mechanisms, and which supports a wide variety of scientific-technical asset classes, is warranted. Supporting the private sector is likely to pay off manifold higher benefits than enduring any socio-economic havoc caused by a pandemic. However, developing vaccines against unknown agents remains risky and is capital and resource intensive. The mRNA experience has also proven that relying on scientific breakthroughs requires high-tech facilities and know-how.

### 3.2. Centralized Procurement Coalitions

#### 3.2.1. Monopsony vs. Monopoly

As vaccines are preventative, the primary customer is the public sector, even more so in times of pandemic crises. If purchasing is organized centrally, the buying authority, as the sole buyer, may exert “monopsony” powers. This puts the government as the buyer in a strong position while negotiating the purchase conditions for public health goods. However, if a government-monopsonist presses prices downward too vigorously, they may well curtail or eliminate the sources of input, by driving any potential suppliers out of the market. In the case of COVID-19 vaccines, the supply at the outset remained limited, before several companies succeeded and manufacturers were able to scale up and produce billions of doses. Consequently, this gave those winning the race “monopolistic” power. This gradually disappeared once more suppliers brought their products to market. As a result, a transactional “prisoners dilemma” occurs, forcing the buyer and seller to reach a negotiated solution. However, buyers bargaining for lower prices in the moment that suppliers are able to set a monopolistic (higher) price, may actually result in a more balanced transaction outcome, benefiting both parties.

#### 3.2.2. Centralizing Procurement

Centralizing procurement provides buyers the advantage of scale and increases their negotiation power, in order to obtain price reductions for pooled volumes. Nevertheless this process is complicated and takes time. Partners tend to differ in their preferences with regard to the type of vaccine for primary series or boosters, vaccination policy and priority groups, budget constraints, and willingness to pay. Achieving consensus depends on the number of partners. However, if procurement alliances are set up, preferably within the same jurisdiction, the benefit of pooled procurement between nations outweighs the challenges encountered for the period of cooperation, which may continue during the course of the pandemic.

At a policy meeting on 12 June 2020, in Brussels, the Council of EU Ministers for Health agreed on the need for joint action to support the development, but also the deployment, of a safe and effective vaccine against COVID-19, by securing rapid, sufficient, and equitable supplies for Member States [8]. In order to implement this action, the Commission offered to run a single central procurement procedure on behalf of all Member States, with a view to signing EU-level so-called “advance purchase agreements” (APA) for COVID-19 vaccines with a number of vaccine manufacturers. The EU APA agreements included upfront European financing to “de-risk” investments, in order to increase the speed and scale of successfully manufacturing effective vaccines. In return, the APA would provide the right, or under specific circumstances the obligation, to participating Member States, to buy a specific number of vaccine doses within a given timeframe and at a given price.

#### 3.2.3. Advance Purchase Agreement

Forecasting vaccine demand when facing outbreaks, or anticipating a full blown pandemic, is by definition difficult. When it happens, any further transmission across the population is hard to predict, while epidemiology-based mathematical models are only valid when the reproduction factor for a particular mutant has been established based on empirical findings and field reporting. Alternatively, stockpiling can be a useful strategy. In the case of outbreaks and pandemics, a strategy based on a priori stockpiling is not possible if the infectious agent is unknown. In the case of COVID-19, advance purchase agreements were intended to solidify the government’s commitment towards addressing outbreaks in the face of uncertainty and served as a risk-sharing policy if governments decided to cancel orders.

On the subject of vaccine demand forecasting, confusion often arises over the definition of demand. The public sector has repeatedly confused need with effective demand [9]. Public health experts define “need” as the number of doses that should be used in all populations to control the disease. However, “demand” is the number of doses that will actually be purchased, depending on a population-wide strategy or vaccination of selected groups, government willingness and ability to pay, and the expression of urgency and readiness.

#### 3.2.4. Nationalistic Appropriation

During the COVID-19 and previous pandemics, the supply of certified vaccines was initially limited, even in affluent countries. As a result, the world has at times of pandemics witnessed trade fights between vaccine-producing, industrialized countries and the developing countries, and governments have threatened each other with sanctions including stopping the export of critical materials and equipment, in order to obtain vaccine supplies on time, or sooner [10]. There is evidence that an unequal vaccination roll-out within and across countries will deepen inequalities. Across all countries, those with the greatest supply shortage were low- and middle-income countries. Almost all countries refer to proportionality, even those who submitted their purchase orders much later. Nonetheless, the low- and middle-income countries have seen their access to supplies being depleted or delayed by wealthier counterparts. Waiting times in the lower-income countries have risen to several months and are expected to last throughout 2022 and 2023. As a result, healthcare workers and millions of elderly and other high-risk inhabitants of lower income countries have remained long-time unprotected. This has limited the impact of immunization in these settings; therefore, extending the on-going pandemic. This not only increases the death toll, but has imperiled these already fragile health care systems and economies [11].

Vaccine nationalism and hoarding by wealthy nations is not a new phenomenon, however. About ten years ago, a vaccine was developed to combat the pandemic of influenza A virus H1N1, known as swine flu. At that time, affluent countries bought up virtually all of the available supplies of that particular anti-pandemic vaccine [12]. In retaliation, developing countries refused to share critical virological samples needed to design an effective vaccine and start its production. Thus vaccine nationalism creates a win-lose situation.

#### 3.2.5. Key Learnings

Government procurement often entails a multiple-bid process, allowing governments to select those manufacturers willing to produce a sufficient volume at the most favorable price. The winning contract usually determines all phases of production and logistics and distribution. As the sole buyer for the nation in times of pandemics, this places governmental buyers in a strong negotiation position. Their monopsony power becomes even greater if several countries decide to pool resources. A drawback of pooling is that this may lead to coordination problems, as there are divergent public health perspectives. Another challenge is to manage nationalistic feelings, so that they do not imperil the wider public good.

### 3.3. Supply Tendering, Quality, and Price

#### 3.3.1. HTA Value Assessment

Cost–benefit and cost-effectiveness analyses are standard procedures when weighing the “value for money proposition” in pricing and reimbursement negotiations or tenders. The price of new vaccines is increasingly benchmarked against healthcare savings and the value to society. Due to their impact on individual, as well as population, immunity and the capability of actually preventing or possibly eradicating infectious diseases, vaccines offer the most influential externalities, by reducing transmission and eventually reaching herd immunity.

In the case of vaccines that can be used in a pandemic situation, there has been debate about whether such a socio-economic evaluation, named “health technology assessment” (HTA), is appropriate in this particular situation. Owing to the presumed high benefit to price ratio for pandemic vaccines, the willingness-to-pay is likely to be greater than in endemic situations; and given the speed of pandemic scaling, decisions by public authorities have to be made urgently and society has actually little choice but to accept the costs [13].

Industry has been very much aware of its corporate social responsibility, and if vaccine pricing had been perceived as “excessive” (although that is legally hard to prove), there could have been government measures or price gouging lawsuits based on competition law infringement. However, a competitive price mechanism is needed to keep price hiking at bay.

#### 3.3.2. The Benefit of Competition

In the global vaccine market, the objective of international competitive bidding, or open tendering, is to provide eligible prospective bidders with an equal opportunity to participate in the tender procurement system. In the majority of cases, awards are based on the most attractive offer from suppliers who can meet the tender provisions in terms of quality, price, and additional supplier services. This includes speed, cold chain logistics, and supply security. A prerequisite is that eligible vaccines need to have been approved by the respective (inter)national regulators; i.e., the Food and Drug Administration (FDA/US), European Medicines Agency (EMA/European Union), or the WHO prequalification system.

The effect of competition on lowering prices can be illustrated by the classical supply and demand curve, which demonstrates the effect of multiple contestants vying for market share. More suppliers will push the supply curve to the right. Due to the increased supply, the market equilibrium price (Y-axis) will decrease, while the number of doses increases (X-axis). In short, the consumer surplus increases, while the supplier’s surplus decreases. The fraction of total welfare benefiting government-buyers (consumer surplus) becomes larger.

Therefore, competition is an important element in price setting and naturally leads to price decreases. However, in the event of breakthrough products, competition initially has little effect in terms of controlling prices. This changes when more competitors come to market. In addition, government subsidy or market driven push and pull incentive systems, as discussed in Section 3.1.2, accelerate the competitive process, with some countries applying both.

In summary, the more competitors enter a competitive market, the more buyers and patients will have a choice. Due to the invisible hand of competition, there will be downward pressure on prices. In this contestable ecosystem, large incumbent firms can be challenged by new radical ideas that destroy the incumbent position (e.g., mRNA). The incumbents may also have a greater incentive to develop riskier technologies; either alone or through partnerships.

#### 3.3.3. Managing Supply Security

Price is not the only benchmark, and procurement agencies select several suppliers utilizing a number of quality and security qualifications. Successful implementation of COVID-19 vaccination programs must also ensure effective vaccine storage, handling, and stock management; rigorous temperature controls across the supply chain; and logistics information systems. This has been vital to safeguarding COVID-19 vaccine supply and to preventing interruptions, from the point of manufacturing through to the vaccination service delivery.

Advanced technology platforms for COVID-19 vaccines have been developed or are currently in the later phases of research and development, each targeting different cold chain requirements for logistics and storage, so as to address the supply chain challenges many countries face. At the moment, mRNA vaccines use a new technology, i.e., strands of messenger RNA (mRNA), which are held in lipid particle envelopes that are vulnerable to degradation at room temperature. This requires the vaccine doses to be frozen for transportation, then thawed for use. That is where Moderna’s vaccine may have a competitive edge. Unlike Pfizer’s and BioNTech’s offerings, it does not have to be stored at −70 °C, but can tolerate a less cold −20 °C, which temperature is standard for most hospital and pharmacy freezers. This difference means that Moderna’s vaccine should be easier to distribute and store, particularly in rural areas in the US and also in developing countries that lack ultra-cold freezers [14]. Furthermore, vaccine supply channels in countries must be secured against potential theft, and anti-corruption measures must also be taken [15].

#### 3.3.4. Procurement Transparency

Transparency during the negotiation process about prices has been a matter of debate for many years [16]. For instance, during the COVID-19 pandemic, the executive branch of government in the US was mandated to negotiate prices and contractual terms directly with vaccine firms. In other countries, this was done similarly, either at a domestic bilateral level, or in a coalition with fellow nations. The latter being the case for the European Union.

Regarding transparency, Members of the European Parliament (MEPs) praised the European Commission’s “joint purchase of vaccines”, which led to a stronger negotiation position than individual EU countries would have had: “That means more vaccines for a better price and under better conditions” [17]. However, the legislative branch wanted to keep oversight and, therefore, demanded contractual transparency. The Health Commissioner, Stella Kyriakides, assured MEPs that their calls for transparency had been heard, and she welcomed the fact that the first vaccine suppliers had agreed to make the text of their contract available. She stated that the EC was working to convince other manufacturers to do the same.

#### 3.3.5. Key Learnings

From a competition perspective, the vaccine sector can be characterized as an oligopoly operating at a worldwide scale. The major contenders are: GSK, Sanofi Pasteur, Pfizer, Johnson & Johnson, Merck, and the Serum Institute of India; which together held over 90% of the market in the pre-COVID-19 era. However, with the exception of Pfizer, which partnered with BioNTech, the major incumbent vaccine developers were late or unsuccessful movers in terms of COVID-19 vaccine development. This proves that the decision by certain governments to broaden the number of competitors, by attracting and incentivizing new entrants, was the right choice. Small and midsize biotechnology enterprises (SMEs) that were previously positioned near the market fringes were now given a chance to fully participate. This turned out to be a major success in accelerating the development of COVID-19 vaccines.

### 3.4. Global Accessibility and Affordability

#### 3.4.1. Gavi International Alliance

The COVID-19 vaccine rollout has been unlike any other prior vaccine delivery effort. The sheer volume of people, doses, and human resources involved required a mammoth global vaccination program [18]. It challenged the most resilient of systems, and even the best-laid vaccine deployment plans were seriously tested. The challenge has been greater when operating in low- and middle-income countries, where inequality gaps are greater and populations are often hard to reach.

Gavi, The Vaccine Alliance, was created in 2000, to support vaccination programs for children in developing countries; particularly, the poorest areas. Gavi developed the Advance Market Commitment (AMC); an innovative program that consists of public–private health funding, of which the objective is to provide an affordable and stable supply of high-quality vaccines at a steeply discounted price and funded by high-income countries. In collaboration with the newly created COVAX facility, Gavi stimulated the development and expansion of manufacturing capacity for vaccines, to support the developing world. Moreover, supplementary support for healthcare capacity building has ensured that the most vulnerable citizens in poorer countries are protected against COVID-19, regardless of income.

The vast majority of companies have pledged support to this Gavi–Covax initiative and arranged to have their COVID-19 vaccine included in the immunization programs of developing countries, and this occurred within one year of introduction in the US and in the EU. Despite this gap in global availability, this nevertheless sets a historic precedent, given the usual average three to five year time lag for introducing new vaccines into developing countries after their introduction in industrialized nations. In an ambitious move to widen the international access to its medical products, including COVID-19 vaccines and antiviral pills, Pfizer announced at a recently held World Economic Forum in Davos that it will provide nearly two dozen of its patent-protected medicines and vaccines, which are momentarily available in the US or EU, at not-for-profit prices to 45 low- and lower-middle income countries. These countries are predominantly located in Africa, Asia, and Latin America [19].

#### 3.4.2. Differential Pricing Policy

Worldwide differential, or tiered, pricing can help boost equitable access to life-saving vaccines. Tiered pricing is considered a pro-access and a pro-innovation mechanism. It allows manufacturers to provide voluntary discounts to poor countries and has no bearing on TRIPS patent law. As will be discussed later, other systems put pressure on the balance between access to medicines and intellectual property and impede property protection.

Thus, a differential pricing policy denotes the practice of firms charging different prices for the same product to different classes of purchasers. This socio-economic pricing mechanism has been described elsewhere [20]. In the context of COVID-19, studies have proposed adopting such a global framework that enables striking a balance between fair access and profits [21]. Tiered pricing makes COVID-19 vaccines instantly cheaper, ensuring large segments of emerging economies can afford them and are not priced out of the market. Along with tiered pricing, Gavi, The Vaccine Alliance, uses a centralized procurement system with a closed distribution network. This makes it impossible for parallel traders to buy Gavi-sponsored vaccines cheaply and resell them in the higher-income segments.

Companies have pledged to provide Gavi-eligible countries with COVID-19 vaccines at the lowest price. However, how to control this promise? There is extensive literature published on whether alleged excessive pricing of pharmaceuticals and vaccines can be identified and corrected through regulation, or by applying competition law [22]. In the case of new, breakthrough medicines, however, it is hard to measure risk-adjusted product development and manufacturing costs and standard accounting rules provide little guidance in retrospect. Nevertheless, under a differential pricing system, the cost of R&D is allocated to affluent countries, while Gavi-eligible countries can be served at marginal production cost. This means producers can operate at full production scale and citizens benefit from the economies of scale. History has also shown that in a special purpose procurement and distribution circuit, such as the one being used by the Gavi organization, manufactures will compete fiercely at the lowest price possible. Moreover, the private sector takes pride in being part of this global partnership. It is considered to be a moral obligation and part of corporate social responsibility.

#### 3.4.3. Scaling Up Manufacturing

Advance purchase agreements (APAs) are warrants that enable accelerating and building manufacturing capacity. During the COVID-19 pandemic APAs had an unintended consequence for high-income countries—as primary public funders of R&D—of tying up most of the supply of doses produced. As Gavi-Covax was unable to compete on equal footing and some rich nations delayed dose donations because of the crisis rampant in their own countries, the world’s poorest countries were forced to wait. This geographic crowding-out effect resulted that allocation based on a forecast was no longer under the producer’s control.

This is illustrated by data supplied by Airfinity (a predictive science intelligence company). The data explain a number of developments: (1) production of vaccines was started before the new compounds obtained market authorization; hence, speeding up the delivery process, but carrying a financial risk of already building capacity while not being certain of approval; (2) the cumulative global production had already reached 11 billion doses by the end of 2021; (3) therefore, as of 2022, the worldwide production capacity should no longer be a constraint and demand instead of supply should determine vaccination rates [23]. Currently, it is the absorptive capacity of the healthcare systems in developing countries that is on a critical path.

The data imply that in the course of 2022, there could even be an oversupply of vaccines, and thus further expanding production capacity may become unproductive. The construction of supplementary manufacturing plants may have to be postponed, especially if the pandemic starts waning. Instead, facilitating international trade may optimize the utilization of existing capacity, as countries have varying supply needs over time and by geography.

#### 3.4.4. Industrial Policy and Health

There is unsurprisingly a political agenda that influences, not only a country’s health policy, but also its industrial biomedical ambitions. Cooperation is needed among different stakeholders at both the domestic and international level, taking into consideration mutual and competing interests. Public–private and private–private partnerships are essential.

SME start-up companies such as Moderna and BioNTech have licensed the University of Pennsylvania’s basic patent, which contains a “US government rights” clause, stating that the patent is “expressly subject to all applicable United States government rights, including any applicable requirement that products, which result from such intellectual property and are sold in the United States, must be substantially manufactured in the United States”. These patents may cover not only the mRNA technology or vaccines directly, but also human body delivery systems technology, such as the lipid nanoparticles that carry the mRNA. Moderna, BioNTech, and CureVac all use these nanoparticle technologies and have acquired access through international arrangements and direct, or indirect, sub-licenses.

Solidarity between countries must be a two-way street. Studies have shown that vaccination with mRNA vaccines made by Pfizer-BioNTech and Moderna offers the best protection against hospitalization. However, vaccines developed with older technology by Chinese manufacturers proved fairly effective against the original strain of the SARS-CoV2 virus but have proven much less so against the more recent variants of concern (VOC). However, more than two years into the pandemic, China has not approved these mRNA vaccines and is not open to international trade. The country’s industrial policy dictates the use of locally manufactured (mRNA) vaccines and meanwhile decided to launch a “zero-COVID-19”, containment policy of non-pharmaceutical interventions (NPIs). This includes massive population lockdowns, indirectly affecting the supply of economies worldwide. International observers and key opinion leaders at the World Economic Forum have criticized this approach. It causes psychological and physical human suffering by keeping people confined to their home and indirectly brings the world to the brink of recession [24].

#### 3.4.5. Key Learnings

A policy of differentiated, tiered pricing greatly helps in making vaccines affordable around the world. Gavi has practiced this policy successfully across its vaccination programs since its inception in 2000. Nonetheless, in the case of pandemics, it seems affordability is a necessary, but not a sufficient, condition. Vaccine manufacturers managed to produce hundreds of millions of doses of COVID-19 vaccine in the first few months of 2021 and scaled up production to over 11 billion by the end of 2021. Although this would be enough to immunize the world’s adult population, it does not take into account politically based delays in allocation; such as a majority of doses going to wealthier countries with some governments taking the precautionary measure of ordering doses in excess of their population.

### 3.5. Equitable Allocation and Distribution

#### 3.5.1. A Global Public Good?

Pathogens know no borders, and infectious diseases spread faster than before in an interconnected world; a situation that is exacerbated by global warming. To bring a pandemic under control and eventually to an end, a large share of the world thus needs to be immune to the virus. At the Global Health Summit, May 2021, Leaders of the G20 and others stated that the COVID-19 crisis will not be over until all countries in the world are able to bring the disease under control; and therefore large-scale, global, safe, effective, and equitable vaccination, in combination with other public health measures, constitutes a political priority [25].

There is a discrepancy in the interpretation of the term “global public good”, where it seems to mean different things to different people, and therefore causes misunderstandings among members of the public and private sectors. In the substantial body of academic literature around the theme of globalization and health, the general aim is to create a better world, where people work together for the good of society without discrimination against race or gender. This is a laudable goal and the principle that G7 and G20 leaders agreed to endorse.

The practice of immunization is a global public good, saving millions of lives every year. However, it is difficult to translate the mission of universal vaccination into tangible outcomes, where achieving equity in allocation of vaccines across regions has proven to be challenging. Although the side effects can never be to discriminate, there arise situations where prioritization of target countries and populations will be inevitable. In that context, the term “public good” will have to be tested against an economist’s definition, of a good that is available for use by all (it is non-excludable) and will not diminish when consumed (it is non-depletable). Clean air is often quoted as a public good that produces a positive externality for society; while pollution, as the antipode, has a negative externality effect on society. Unfortunately, vaccines are depletable and are thus not a public good due to their initial scarcity.

Who will make the hard choice of allocating vaccines and set priorities for worldwide distribution?

#### 3.5.2. Pandemic Governance

At the beginning of the COVID-19 pandemic, an ambitious plan was conceived aimed at global governance of COVID-19 vaccines. The COVID-19 Vaccines Global Access, abbreviated as Covax, was set up as an NGO initiative aimed at equitable access to COVID-19 vaccines. It is co-lead by the Gavi Vaccine Alliance, the Coalition for Epidemic Preparedness Innovations (Cepi), and World Health Organization (WHO), alongside the delivery partner UNICEF.

At the beginning, it was conceived of as an end-to-end program, spanning new vaccine development to healthcare delivery, and for every country in the world. By investing in several COVID-19 vaccine candidates from different manufacturers, Covax would improve the chances of having an effective vaccine for successful roll-out right after the approval of the respective market authorization authorities. Buying doses in bulk would mean that Covax would be in a strong position to negotiate with suppliers and obtain favorable prices. Covax would operate at a global level and serve everyone’s needs, from one central location rather than working in a decentralized manner through mobilizing country or regional authorities.

Covax’s hope was that the high- and middle-income countries would buy from Covax, while poorer countries would receive vaccines almost for free, funded through donations from wealthy governments and charities, and for up to 20% of their population. By operating as a clearing house, Covax would allocate the available vaccines fairly across the world, shipping them to rich and poor countries alike. Sadly, the Covax facility as a concept did not deliver on its promises. Currently, Covax (together with The Gavi Alliance) has stepped back and decided to focus on its core mission of serving the poorer countries.

This “one-stop” initiative was said to have been too optimistic in its expectations [26]. It did not take into account the diversity in buyers and planned activities by the high-income nations in order to secure vaccine doses for their constituents through advance purchase contracts. Hence, advocates and government officials of low- and middle-income countries came to the conclusion that uneven distribution of vaccines was not going to be fixed by the Covax purchasing system and that regulating any global allocation requires a different system.

#### 3.5.3. Influenza Pandemics

This has not been the world’s first pandemic. Can we learn from the past? Vaccine nationalism dominated pandemics before, apparently without lessons being learned. During the past influenza pandemics, about ten years ago, the US pledged to donate 10% of its vaccine purchases to the WHO. Yet only a few months thereafter, the US Secretary of Health and Human Services stated that the US would actually not donate part of their H1N1 vaccine as promised, until all at-risk Americans had had access, since at that time production problems had created shortages [12]. A similar scenario of non-sharing arose during the COVID-19 pandemic.

The consequences are dire. The lack of a mechanism to ensure equitable access to vaccines during the influenza pandemic prompted Indonesia to refuse to share H5N1 virus samples with the WHO that would have been used for pandemic surveillance and tracking viral spreading [27]. The WHO entered talks with governments, in order to secure a number of vaccines for developing countries, and appealed for monetary donations to purchase vaccines and supplies to help developing countries address the 2009 H1N1 pandemic. These diplomatic efforts yielded donation pledges from higher-income countries and manufacturers. Nevertheless, these donations in themselves proved not to be sufficient to address the shortages in supply in developing countries compared to their industrialized counterparts.

#### 3.5.4. Allocation Inequities

At present, there still is a discrepancy in the vaccination coverage rates between high-income, higher-middle-income, lower-middle-income, and low-income countries (following the World Bank classification based on gross national income) [28]. As seen in previous pandemics, the wealthier countries have hoarded vaccines and claimed sovereignty. They use a hedging strategy, by ordering doses in excess of their population and keeping their options open.

As shown in Section 3.4.3, any supply constraints seen in the first year of the COVID-19 pandemic have been resolved. This shifts the burden from the supply to the demand side. The low- and middle-income countries will have to start catch-up vaccination campaigns, to close the existing gap in vaccination coverage rates. Their national health care systems, therefore, need strengthening, special support, and funding. The Gavi Vaccine Alliance has experience in providing such support to poorer countries, in order to achieve equitable and sustainable childhood vaccination. This requires capacity building in procurement, cold chain logistics, and training of healthcare workers. The COVID-19 pandemic requires comprehensive vaccination programs that reach beyond childhood age groups, because the burden of disease is more prevalent among adolescents, adults, and the elderly.

Due to the improved supply situation of COVID-19 vaccines, the EU has decided to expand its international efforts, not only by supporting the efficient use of available doses, but also to step up extra funding of health services to accelerate vaccine uptake in developing countries. Dr Ursula von der Leyen, President of the EC, declared on 12 May 2022, “The supply of vaccines must go hand in hand with a speedy delivery, especially in Africa. The priority today is to make sure that every dose available is administered. And because we know that the best answer to any potential future health crisis is prevention, we are also stepping up support to strengthen health systems and preparedness capacities” [29].

#### 3.5.5. Key Learnings

When vaccines become available, prominent ethical issues around their equitable allocation and distribution arise. Public health agencies, at national and regional level, are dealing with fair triage against a backdrop of a limited availability of doses in the beginning. Prioritizing access for specific population groups will necessarily delay access for others. As seen in previous pandemics, the wealthier countries start hoarding vaccine doses by claiming state sovereignty. The question, therefore, is who has the authority and responsibility of prioritizing vaccine allocation across the world? One can argue that governments are mandated to focus primarily on the well-being of their constituency. In the case of pandemics, however, the counterargument may be that global stability is equally important and can be achieved by promoting the greater public good. Hence an adequate governance system must involve the highest executive branches that represent national governments.

### 3.6. Compulsory vs. Voluntary Licensing

#### 3.6.1. Are Patents an Impediment?

In accordance with government regulation, a patent protects the “exclusive right” of the holder, during the legal patent period from “copycats”. There are however two things that are often misconstrued. First, other R&D-based firms that have developed their own proprietary technology have the authorization to freely compete. Second, to obtain a patent right, the innovator is required to disclose detailed proprietary information that allows the patent authorities to verify whether the claimed invention is indeed novel, non-obvious, and useful. By doing so, patenting has a public disclosure function that allows other innovators to build further on any previously disclosed inventions. The alternative is to dramatically abolish intellectual property rights and for the innovators to keep the information secret.

Essentially, the patent system allows innovators to exercise their rights and recoup any risky investments that have been made. Nonetheless, the patent system works best for medicines and vaccines against diseases that primarily afflict the more affluent high and middle-income countries, where the ability and willingness to pay are adequate proxies for the societal value of the innovation. This is not entirely the case in preventing COVID-19, by which poorer countries are equally affected. There is a potential risk that prices charged during the patented period will be considered “too high”. It is therefore important that throughout the pandemic, vaccine manufacturers pledge to keep prices of COVID-19 vaccines significantly lower than other routinely administered vaccines; apply tiered pricing; and in particular, to price vaccines at cost within poorer nations. In other words, not enforce their patent rights in developing countries during pandemics. This is not the same as requiring they give up patent rights and that third parties may duplicate without consent.

#### 3.6.2. Compulsory Patent Licensing

Alternatively, compulsory patent licensing has been set out in Article 31 of the Trade Related aspects of the Intellectual Property rights agreement (hereinafter: ‘TRIPS Agreement’) and in Article 5A (2) and (4) of the Treaty of Paris. The TRIPS Agreement is an annex to the WTO Agreement that makes it mandatory for each signatory to provide patent protection for medicinal products; an important warrant for innovation in R&D based industries.

Article 31 of the TRIPS Agreement stipulates that compulsory licenses (which are not explicitly mentioned as such) may only be permitted, if prior to such use, the proposed user has made efforts to obtain authorization from the patent right holder on reasonable commercial terms and conditions, but that such efforts have not been successful within a reasonable period of time. This requirement can be waived by WTO members in case of a national emergency or other circumstances of extreme urgency, or in cases of public non-commercial use. Moreover, the scope and duration of use shall be limited to the purpose for which it was authorized [30]. During the COVID-19 pandemic, the EC’s position has long been that, (only) if voluntary cooperation fails, compulsory licensing within the WTO’s existing Agreement on Trade-Related Aspects of Intellectual Property Rights should be used to facilitate the expansion of production and sharing of expertise. However, some countries found the TRIPS procedure burdensome and started lobbying for a more radical approach.

#### 3.6.3. Intellectual Property Waivers

South Africa and India were the first countries that called for extreme relaxation of patent protection rules, and since then their proposal to temporarily suspend intellectual property (IP) rights concerning COVID-19 vaccines has been heavily discussed at the WTO. Such a patent waiver would allow developing economies to manufacture the new mRNA vaccines without acknowledging patent rights, nor would there be an obligation to consult, obtain agreement from, or pay the originator company, as is done under compulsory licensing.

We concur with IP life science experts, including the Max Planck Institute for Innovation and Competition Law, that the patent waiver initiative is excessive [31,32]. A crucial factor in this approach is that new vaccines, such as those of BioNTech/Pfizer and Moderna and, if authorized in the future, of CureVac, are based on messenger RNA derived from science-based technologies that are protected by basic patents that have already been granted or are still to be granted. The associated medical technologies have other very promising areas of application, namely in cancer therapy. If the patent protection for vaccines were to be suspended, this would have repercussions on applications in, for example, therapeutic domains beyond preventative applications. Abolishing patent rights will work as a disincentive for biopharma companies to further invest in innovative technologies. Interestingly, South Africa meanwhile has signed a deal with a local biotech company Afrigen, with the full support of WHO’s Director-General Dr Tedros Ghebreyesus. Despite “copying” Moderna’s technology, the consortium acknowledges that it cannot deliver its vaccine in the coming years to fight the pandemic [33]. Moreover, it remains to be seen whether South Africa can and will supply other nations in Africa on an equal footing. It will likely face the same allocation problems. India already decided to stockpile its vaccine for local usage only and has blocked exports [34]. After the decision of South Africa, in collaboration with WHO’s technical unit, to go ahead with “reverse engineering” the Moderna vaccine without the original manufacturer’s consent, the latter then decided to invest in building a production facility in Kenya in order to produce a range of its mRNA vaccines, and not only the COVID-19 pandemic vaccine, by signing a long-term collaboration agreement [35].

#### 3.6.4. Voluntary Licensing Practice

The manufacturing and supply of vaccines can be scaled up so that the worldwide access is expedited without waiving a vaccine’s IP, but through voluntary licensing; a scheme in which patent owners give other producers permission to use their patented products under favorable terms, to expedite access in lower income countries, subject to certain quality conditions.

Voluntary licensing is nowadays standard practice in biomedical technology transfer and is extensively applied between pharmaceutical and biotechnology firms for both R&D and manufacturing purposes. For example, Moderna has engaged in a number of strategic licensing agreements, including with other manufacturers; e.g., Merck and AstraZeneca. Furthermore, for the purpose of significantly amplifying the manufacturing capacity of its COVID-19 vaccine, Moderna entered into third party agreements with contract manufacturers. In May 2020, it signed a long-term agreement with Lonza, a Swiss multinational firm that provides development and manufacturing services to the biopharma sector. Lonza is a major partner to innovators, with production facilities in Europe, the US, and Asia.

Without the voluntary cooperation of the originator company, a significant and rapid increase in production capacity is unlikely to happen. It is unlikely that a company can be forced to hand over its “know-how” by way of a TRIPS waiver. In contrast, voluntary licensing has a higher chance of success and is based on a close collaboration between the licensing parties. This involves dedicated teams of scientists and engineers working together.

Patents do not automatically include the “secret sauce” for the production of high-tech products, particularly in the field of complex biological medicines, such as vaccines [36]. This is not because the patent applicant wishes to keep the information secret, in many cases the processes and systems are too complex to provide a written description in detail. Even with full support of the original manufacturer that is willing to provide the technical know-how regarding materials and processes for the production of biological medicines, it remains a lengthy and challenging operation for a third-party licensing collaborator. Last but not least, granting a patent license not only protects against uncontrolled imitation but provides the legal certainty that licensees will use the technology in accordance with the contractual terms and apply the original company’s specifications and quality control.

#### 3.6.5. Key Learnings

Neither patent waivers nor compulsory licensing are essential in achieving access and affordability. The critical issue is the scaling of production and, in particular, the “fill and finish” at the end of the process. If technology transfer is needed, this is best realized in mutual agreement, and voluntary licensing is the standard procedure between manufacturers. Abolishing intellectual property would be counterproductive and discourage investment in future R&D. After COVID-19, there will be many more epidemics, due to climate change and global warming. A new quest has already started for a broadly-protective anti-coronavirus vaccine, and a multivalent vaccine in combination with an influenza vaccine [37]. Covering multiple strains reduces the number of doses administered and improves patient compliance, hence increasing vaccination coverage rates. Other desiderata are heat stability and convenient administration modes such as oral administration or a nasal spray.

### 3.7. Toward a Universal Pandemic Treaty

#### 3.7.1. Pandemic Preparedness

No one can say whether the next pandemic will be caused by a coronavirus, influenza, or another, but with better public surveillance systems and laboratory diagnostics we should in principle be able to rapidly identify such a new pathogen and devise countermeasures before the outbreak has a chance to spiral out of control. This requires public health measures and preparedness, before the pandemic takes place. There is “no time to lose” [38].

A broader strategy that goes beyond the tasks of early detection and devising a new technology solution is needed. Global and local healthcare systems must be strengthened, along with organizing pandemic-related preparedness training, which needs to be maintained during the inter-pandemic periods. There is experience with this matter. During the last avian influenza pandemics, the Public Health division of the EC used to run cross-border strategy training in Luxembourg and Brussels. In addition, virtually every Member State ran exercises. However, more can be done at the national and international level and by involving key stakeholders. The Gates Foundation calls for creating a global pandemic preparedness “fire brigade” comprised of three thousand experts scattered around the world, recruited for skills ranging from epidemiology and genetics, through vaccine and medicine development, and computer forecasting and intervention modelling, to diplomacy [39]. Such a type of organization, which would probably best work under the auspices of the World Health Organization, would remain on permanent standby, ready to respond to any detected outbreak, and such preparedness is estimated to cost USD 1 billion a year.

An important task is addressing vaccine hesitancy. Vaccination requires at least two to three doctor visits over a period of one or more years and (annual) booster shots are expected to be needed [40]. This makes compliance and completion of the entire course a real challenge, as the (potential) pool of vaccinees must be reminded and followed up. This requires setting up a lay-public orientated, communication and education program network. Moreover, experience with the current COVID-19 pandemic has shown that vaccine hesitancy and conspiracy theories present a major impediment to the management and control of pandemics. Widespread education programs through medical channels must be accompanied by confronting the spread of misinformation on social media that may include conspiracy theories that sow doubt about the effectiveness of certified vaccines, as well as non-pharmaceutical interventions (NPI), such as lockdowns and mask mandates.

#### 3.7.2. Global Health and Politics

In many high-income countries, preventing, detecting, and responding to disease outbreaks is rooted in a state-centric vision of national security [41,42,43]. By engaging with global pandemic preparedness and response organizations, governments can protect their populations and economies from infectious disease threats. During the COVID-19 pandemic, the world witnessed the extent of this self-centricity: from the very early stages of the pandemic there was a retrenchment from a shared global vision of support for those most in need to a nation-state first approach, which was termed “vaccine nationalism” (see Section 3.2.4).

As is clear from the different government responses to the COVID-19 outbreak, health policy decisions involve political decisions about who should be consulted, who should provide advice, which models should be used, what strategies should be implemented, how these policies should be enforced, and who should be trusted in the international arena. Hence political tensions are not limited to the domestic arena [44]. International organizations such as the WHO operate in an increasingly combative and divisive political environment, with proxy battles being waged between member states of these institutions, as observed between China and the US, and, in particular, during the Trump presidency.

#### 3.7.3. Universal Pandemic Treaty

In the spirit of pandemic preparedness, the WHO, as a global health agency with an international team of experts, plays a crucial role. Although the WHO certainly deserves merit in ensuring pandemic preparedness and generating surveillance data, it faces its limits when it comes to setting priorities in the allocation of vaccines during a pandemic.

At a Special Session of the World Health Assembly (WHASS) in 2021 [45], WHO Member States agreed by consensus to establish an intergovernmental negotiating body (INB) to draft a convention, or agreement or other international instrument for pandemic preparedness and response under the Constitution of the World Health Organization. The underlying logic of such a “pandemic treaty” is the awareness that governance failed during COVID-19. A treaty is vital to add political commitment to the technical knowhow, recognizing the lack of cooperation and coordination between governments, and member states being overly focused on national protectionism, causing inequities in the vaccine allocation.

Proponents argue that a global pandemic preparedness treaty, rooted in the “norms of solidarity, fairness, transparency, inclusiveness and equity”, would be the cornerstone of future global health security and that this would overcome many of the shortcomings seen in the response to COVID-19. On the other hand, international affairs experts argue that there is a mismatch between the national self-centric problems observed during the response to the COVID-19 pandemic and the envisioned pandemic treaty, which proposes to take a globalist, quasi-cosmopolitan approach to pandemic preparedness and response: “Championing solidarity and equity requires states to depart from state-centric policy-making and focus on the global, something states have been unable or unwilling to do in global health governance to date and indeed during the COVID-19 pandemic” [46]. To move forward, they argue that rich countries must answer the question of “what they are willing to give up” nationally, in order to be prepared internationally and be able to fight future disease pandemics.

#### 3.7.4. World Health Organization

For a pandemic treaty to have teeth, the entity that governs it needs to have power, whether political or legal, in order to be able to eventually enforce compliance. In its current form, however, the WHO does not possess these powers. The manner in which states departed from WHO guidance during COVID-19 demonstrated that there is a huge problem with the legitimacy of the WHO in the eyes of the global community. Consequently, Schwalbe and Lehtimaki [47] stated that “a treaty negotiated under the auspices of the WHO, which has little authority of its own and instead reflects the interests of its member states, will be unlikely to make the sweeping changes that are urgently needed”. The WHO takes a globalist, cosmopolitan world-view, its mandate asserting that the “highest attainable standard of health is one of the fundamental rights of every human being”. However, for this treaty to work, it needs to be able to hold the members to account for nationalistic behavior.

At the moment, however, it seems unlikely from a political perspective that all of the WHO’s member states will be willing to give this globalist institution more power to hold its members accountable, even though this might make the world better prepared for the next pandemic.

#### 3.7.5. Key Learnings

To accomplish the global equitable allocation of pandemic vaccines and resolve the key issues encountered at cross-border level, a pandemic governance treaty between all nations is required. Therefore, on 1 December 2021, the 194 members of the World Health Organization (WHO) reached a consensus to begin the process of drafting and negotiating a convention, agreement, or other international instrument under the Constitution of the WHO aimed at strengthening pandemic preparedness and prevention through a coordinated response. Many who favor the treaty believe that it also offers the best way to increase political commitment from states, to reform global health governance. However, the experience with COVID-19 revealed that this proposition may not necessarily be realized. The position of member states on the substance of the treaty remains to be seen once a draft is completed.

## 4. Discussion

Many high-income countries, including the US, the UK, and the 27 nations in the EU, have sought to quickly secure reservations of newly created capacity for vaccine production and delivery for their domestic deployment, often in return for sizeable investments in research and development and financing “at-risk” scaling of vaccine manufacturing capacity. These industrialized countries have appropriated vaccine allotments through bilateral agreements, i.e., by concluding advance purchase agreements with the R&D-based pharmaceutical and biotech industry. The latter constitutes good practice and an emergency response.

Unfortunately, these bilateral agreements make an equitable distribution of vaccines at the global level difficult. Vaccine nationalism leads to a win–lose situation and if a bidding war starts, or international trade is severely hampered, this may quickly evolve into a lose–lose situation. Furthermore, if the low and middle-income countries are not timely served, this may prolong the pandemic. Consequently, besides the humanitarian considerations, even from a self-centered perspective, it makes sense for industrialized countries to share doses based on infectious-disease epidemiological needs. However, that did not happen during the critical first and second years of what meanwhile has become a protracted pandemic.

Effective pandemic preparedness means focusing on detecting the first wave and being ready to mobilize worldwide health resources. History has proven that global solidarity and securing access on an equal footing does not happen automatically. To ensure the equitable roll-out of life-saving vaccines, a pandemic pact, or treaty, between nations is necessary. The WHO has agreed to launch this process [48]. The expectation is that such a pact will remedy the lack of sharing of technology, information, resources, and data that were the COVID-19 pandemic’s defining characteristics. The hope is that the treaty will make countries accountable to one another, an ambitious goal! It remains to be seen whether the pact can be enforced and how much power state leaders are willing to hand over to the WHO.

We agree with international affairs experts who suggest that the WHO needs to pay serious attention to the competing diplomatic priorities among its member states. The WHO leadership has used primarily name-and-shame tactics to reproach countries that decided to chart their own pandemic course. WHO’s failure to grasp the political priorities of all member states has left the organization struggling to enforce its authority. Given the fact that infectious disease pandemics have, besides their negative impact on global public health, dire consequences for the economy and security, governance must be such that it actively involves state leaders in decision-making and the ratification of the treaty.

## 5. Conclusions

In this paper we have examined governance and ethics issues, from invention to delivery, through the lens of key stakeholders that together serve the needs of citizens, i.e., the public health authorities at national and international level, governments of industrialized and emerging economies and the poorest countries, and the pharmaceutical and biotech industry comprised of multinational corporations and smaller, entrepreneurial ventures.

Our findings indicated that partnerships are vitally important but must be carefully constructed to achieve optimal effects. Partnership requires mutual trust and transparency, and partners must do their utmost. Within a multi-stakeholder partnership each should focus on and lead the core activity they are best at: i.e., product development and manufacturing, vaccination and public health systems, and inter-governmental coordination and funding.

Product development and manufacturing is the expertise of enterprises that have a track record of delivering existing and new products of high quality in an efficient manner. They are used to working with external partners in the science/technology innovation ecosystem, i.e., to scale laboratory and manufacturing operations, to conduct cross-border clinical trials, and to work with medical regulatory authorities to obtain registration and market authorization. The governments of industrial countries were instrumental early in the COVID-19 pandemic, in creating dedicated push and pull incentive mechanisms that stimulated these functions. This includes upfront R&D grants and advance purchase agreements that come into effect once the vaccine is successfully developed and procured by agencies.

Public health authorities at national and international level develop vaccination policy and set priorities for vaccine allocation and deployment. Since the virus was spreading rapidly across countries, hard choices had to be made about who should be vaccinated first. Within countries, strategic decisions were made as to age groups and at-risk profiles. However, it has proven to be more difficult to perform such triage and priority setting at the wider, international level. In the absence of a global government, health authorities and industries came under political pressure to favor industrialized countries, with the rationale that vaccines were made and financed by those affluent, highly-industrialized nations.

This has left developing economies and particularly the poorer countries in the cold during the first year. However, since then, the production has ramped up quickly. Unfortunately, it is the first year that counts during a pandemic, and despite the extraordinary rapid development of vaccines, not all citizens could be served. Due to the wide impact of such a gap in vaccine allocation and distribution on the world community’s health, economy, and security, this has sparked consternation and a fierce debate around possible solutions. This is not new and happened before with other pandemics such as avian influenza a decade ago.

The crux of the matter is that the overall allocation at the cross-border level is geopolitical in nature and can only be resolved between nations. This can be done bilaterally by sharing doses. However, rather than working through an ad hoc bilateral procedure, it requires a global system that is science-based and which has been decided upon before the next disease pandemic happens. Furthermore, the WHO’s surveillance network needs strengthening, it also requires local health care system preparedness. Rich countries have a moral duty and should help poorer nations, in order to contain any viral mutation spillovers.

A pandemic treaty is indispensable and should be developed under the auspices of the G7 and G20. The aim would be to agree in advance upon scenarios that include fair allocation of scarce vaccines based on public health recommendations. This must contain the proportion pledged to be shared during the course of a pandemic. No matter how fast supply can be scaled through networks, there will always be scarce resources in the first year.

The political debate has shifted, however, and the focus now is on compulsory licensing and waiving the originator company’s patent rights. We argue that this is beside the point. It does not solve the allocation problem, nor does it provide faster supply. IP should not be used as the scapegoat for failures in observed vaccination inequity and the uneven allocation of vaccines among nations. IP is considered the bedrock of biomedical innovation and triggers cooperation between technology partners through licensing. Taking IP away from innovators works as a disincentive, whereas innovation needs incentives, as we have argued.

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
