# Peer review of "The Race for COVID-19 Vaccines: Accelerating Innovation, Fair Allocation and Distribution"

_vaccines, 2022, doi:10.3390/vaccines10091450_

Round 1

Reviewer 1 Report

This article deals with the crucial steps that have accompanied the production of the vaccine.

These phases have had both scientific-technological and ethical-social implications. This last aspect is linked above all to the choice of the allocation of the small number of vaccines available.

Well, the present paper takes its cue from what happened in the Western States during the first wave of the pandemic Covid-19 to analyze the policies related to the financing, production, and subsequent distribution of vaccines.

The following considerations may be made concerning the method of the study.

The authors' method is correct and reproducible, analyzing a wide variety of aspects and points of view.

The results produced by the paper are useful for all stakeholders in the vaccine field. In addition, these results provide a basis for future studies

Author Response

Dear Reviewer,

Thank you for your comments. Based on the various comments received we have incorporated minor changes into the original manuscript and have revised the entire document as to improve legibility.

Reviewer 2 Report

This very long manuscript summarizes the global scenario on vaccine development, procurement and distribution occurred during the COVID-19 pandemic. The arguments appear not sufficiently supported by scientific documentation, generally reporting just the view of the two authors. Please, consider my comments below:

lines 40-42 here the authors express an interest. However, in the introduction, the aim of their work is not stated clearly;

line 47, in Materials & Methods the authors refer to a "comparative analysis", with no references regarding the scientific methods used. Is it a process that come from the opinions of the authors only?

lines 60-65, please note that entire sentences are the same from the paper cited as reference 1;

lines 62-63, as the authors refer to personal ideas, I can state mine here. The safety of this new technology must be proven still. Although very high vaccination rates reached nationally, on January 2022 we registered over 3.8 million positive cases daily worldwide; on March 2022, over 1.8 million positive cases daily; on July 2022, we have still over 1.1 million positive cases daily. Before the vaccination campaign started, we had around 0.6 million positive cases daily. Regarding COVID-19 vaccines' safety, a number of scientific papers report severe under reporting of serious adverse effects, with local studies highlighting a 100 x factor respect to the official passive pharmacovigilance for COVID-19 vaccines.  Please, rephrase lines 62-63 on safety and highly effectiveness of COVID-19 vaccines. 

Lines 98-101 refer to risk mitigation. The statement "in the case of brand-new classes like 100 mRNA has clearly paid off" depend on who is considered for the gain obtained.

Lines 106-115. Front-loaded funding provided by governments to substantially accelerate the new vaccine development process, together with the risk associated with technical failure from the vaccines' companies do guarantee neither the necessary scientific independence nor ethical commitments. 

The section Results covers lines 57-803, I suggest to shorten narrative and inconclusive text.

Line 852. "Mutual trust" would be rather be substituted by transparency.

Lines 855-856. Ethics is completely forgotten among "the core activity they are best at".

Lines 891-893 and 900-902 summarize the sense of the manuscript, I assume.

Overall, the aim of the manuscript is not clear at all, the title is not explicative, methods refer to no scientific methodology, and the results are just narrative - even if interesting - and confounding respect to the conclusions. In my opinion, the scientific quality of this manuscript do not fit with the level of Vaccines. I appreciate the effort of the authors, however I would recommend them to select a couple of aims and focus the manuscript on these arguments. 

Author Response

Dear Reviewer,

Thank you for your comments. As a result, we incorporated a number of changes, as follows:

Manuscript Length:

We agree that the length is above average and it requires readers to digest a lot of information. However, we wanted to write an article that provides an holistic view addressing related issues. We could have focused on one or only a few aspects but that has been done by the independent stakeholders. Only a few publications try to provide a comprehensive strategic and public policy perspective.

Safety and Efficacy:

Upon your indication, we rephrased the lines on safety and highly efficacious Covid-19 vaccines, as follows:  “The development, testing and approval of several vaccines against SARS-CoV-2 in less than one year has been one of the great achievements in society's battle against Covid-19. In retrospect, particularly the mRNA-based vaccines were developed rapidly and have been accepted by EMA and FDA as safe and highly effective, although not against all mutations of the virus.”

Risk Mitigation:

Your remark was that the sentence about investing in new technology like mRNA “has paid off”, is inappropriate. We  did not mean this to be associated to monetary gains but that diversifying investments across various scientific pathways and asset classes “pays off”. To avoid confusion, we modified the text : “ While diversification among classes is no guarantee for potential losses, investing in a broad number of ‘existing classes’ and, in addition, ‘new science’ technology is a prudent strategy; which in the case of brand-new science classes, like mRNA, turned out to be successful.

Front-loaded Funding:

The sentence has been modified and now includes commitments/requirements to receive R&D  funding:

“In sum, front-loaded funding provided by governments (push subsidies) has shown to substantially accelerate the new vaccine development process. Yet, R&D-based companies will still assume the risk associated with technical success, or failure. The advantage of such government funding is that it compensates for the ‘crunch time’ in launching innovations earlier than otherwise would be possible. Accelerating the project development time demands extra resources with regard to skilled labor, infrastructure and ‘at risk’ financing.

At the moment, frontloaded funding does not include ethical and contractual requirements on prioritization and allocation when a vaccine is successfully developed and launched.

Mutual Trust:

We agree that transparency should be added but do not want to give up on the importance of  ‘mutual trust’. In our 30 years’ experience working with senior officials of the public and private sectors we noticed that the ‘lack of trust’ can be a major impediment in building constructive partnerships. It is the aim of articles like this to build bridges between stakeholders by clarifying viewpoints.

The sentence has been modified: “Our findings indicate that partnerships are vitally important but must be carefully constructed to reach optimal effects. Partnership requires mutual trust and transparency and partners must do their outmost. Within a multi-stakeholder partnership each should focus on and lead the core activity they are best at: i.e. product development and manufacturing, vaccination and public health systems, inter-governmental coordination and funding.”

Reviewer 3 Report

I think this manuscript should be a review, not an article. So, Materials Methods section may not be required. Furthermore, the Materials and Methods section does not detail the methods used, such as statistical methods.

Additional literature data can also be cited to support the article conclusions.

I'm rather confused by the type of this article is.   If it is a review, in fact, as long as the author can draw relevant conclusions through sufficient article conclusions. Review manuscript can be accepted or minior edited as long as the focus is highlighted, the structure is reasonable, the data is detailed, and at most some language improvements are made. However, at present, since this article involves many aspects of the new crown vaccine development process, from the perspective of the cited references alone, the number of references is not enough to support so many arguments, and the material methods section should not be needed.  

If it is a research article, there are many areas for improvement. For example, the materials and methods section does not describe the methods used in the article in detail, including statistical methods. Could use a more intuitive way, such as a chart, to show the author's conclusion. If this is the case, I recommend that the essay focus on one point and elaborate on the argument. The current article is written too broadly.

Author Response

Dear Reviewer,

Thank you for your comments.

You say you are confused about this type of article and believe this article is written too broadly. We agree that its length is above average and it requires readers to digest a lot of information. However, we wanted to write an article that provides an holistic view addressing related issues. We could have focused on one or only a few aspects but that has been done by the independent stakeholders. Only a few publications try to provide a comprehensive strategic and public policy perspective.

This does not mean that the manuscript should be considered, or become, a review article that is being guided by a structured literature search. We have done an extensive literature search that comprises about 250 articles of which about one fifth has been retained as references for this article.

You also recommend that the essay should focus on one point and elaborate on the argument. Frankly, this would have made our task more easy. Selecting an appropriate journal would also have been more easy if we had separated the issues in, for example, innovation management, or public health policy, or international affairs and policy. This would have defeated the purpose of educating the various stakeholders into understanding the viewpoint of partners and how to reach consensus.

We ultimately decided to publish in the Journal Vaccines as the focus is on preventing infectious disease pandemics through vaccination. We were particularly interested in being part of the special issue concerning vaccine allocation and ethics. We decided to link this to the innovation  aspects as this would give an overview of the upstream and downstream processes and how they are linked. We have seen so many times how various stakeholders are locked up in their silos. Without the conceptual framework we provide, the saga of mismanaging pandemics may continue. 

Round 2

Reviewer 2 Report

Dear Authors,

Although your aim to provide comprehensive strategic and public policy perspective is understandable and appreciable, both the manuscript length and content are out of the Research manuscript classification as defined in the authors’ instructions (and as intended in any other scientific journal):

Materials and Methods: They should be described with sufficient detail to allow others to replicate and build on published results. New methods and protocols should be described in detail while well-established methods can be briefly described and appropriately cited. Give the name and version of any software used and make clear whether computer code used is available. Include any pre-registration codes.

Results: Provide a concise and precise description of the experimental results, their interpretation as well as the experimental conclusions that can be drawn.

In this manuscript there are neither methods/protocols described, nor experimental results; the content “just” highlight the respectable opinions of the two authors, i.e. this work sounds not “comprehensive” when it comes to safety and efficacy of COVID-19 available vaccines, as many are the papers highlighting new data on that. Besides, many of my previous comments have remained with no answer:

lines 40-42 here the authors express an interest. However, in the introduction, the aim of their work is not stated clearly;

line 47, in Materials & Methods the authors refer to a "comparative analysis", with no references regarding the scientific methods used. Is it a process that come from the opinions of the authors, only?

Lines 106-115. Front-loaded funding provided by governments to substantially accelerate the new vaccine development process, together with the risk associated with technical failure from the vaccines' companies do guarantee neither the necessary scientific independence nor ethical commitments.

The section Results covers lines 57-803, I suggest to shorten narrative and inconclusive text.

Lines 855-856. Ethics is completely forgotten among "the core activity they are best at".

Overall, the length of the manuscript requires readers to digest an exceeding amount of information, the aim of the manuscript is unclear as it focuses on the policy and strategic aspects of accelerating innovation, fair allocation and distribution without a real “comparative” method; indeed, methods refer to no scientific methodology, and the results are quite narrative - even if interesting - and conclusions come from the opinions of the authors, actually. I still believe that the scientific quality of this manuscript does not fit with a “research article” published in Vaccines.

Author Response

Vaccines/ MDPI

Title manuscript (updated by authors on 28 August 2022)

New title:

The Race for Covid-19 Vaccines: Accelerating Innovation, Fair Allocation and Distribution.

Dear Reviewer 2,

We have made improvements in the text of our manuscript; please, find our 3thd updated version in attachment(28 August 2022). We have also taken the opportunity to slightly modify the title; which now reads as: “The Race for Covid-19 Vaccines: Accelerating Innovation, Fair Allocation and Distribution”. This paper is now ready for further processing as far as we are concerned.

We regret that, in contrast to all other academic readers we submitted our paper to, you keep referring to our manuscript as not(entirely) worthy of being considered for publication in Vaccine/MDPI, and that, in particular, you consider the methods and materials used as not ‘scientific’. Based on your comments it is clear to us that you are a proponent of ‘quantitative methods (only)’(which we have used as well in previous research papers, as appropriate). Regrettably, you seem to exclude the other academic pillar of ‘qualitative research methods’ that is considered equally important and valid.

We will explain below our thinking around the major comments that you have made:

  1. Research Methods

Our manuscript is an applied policy research paper, the preferred research method that lends itself to analyzing policy and strategic issues, and that is commonly, if not exclusively used, in academia and policy-related journals.

Such academic research is predominantly descriptive and qualitative in nature. This is in contrast to applying quantitative methods which are commonly used to analyze data and results obtained through experimentation as acquired in either a laboratory setting, or by externally conducting, for example, clinical (field) trials, in a particular setting. In the latter case, statistical validation methods are applied. The ‘scientific method’ you refer to requires that such an experiment can then be repeated by any third party which, if found valid, should lead to a similar result. But such requirements cannot be fulfilled in public policy and strategy research.

We suspect that some of your readers may struggle with this in a similar way, and we advocate to keep the Materials and Methods section which we have updated and expanded to make things clearer to, for instance, readers with a clinical research background. Other journals have opted to create a ‘policy section’ to resolve confusion what this journal may consider doing.

The Materials and Methods section in our updated document(28 August 2022) now reads as:

“This is an applied policy research article. It offers an inter-disciplinary framework for pandemic-related vaccine development, deployment and governance. We use descriptive methodology to identify a number of key issues that have been encountered across the time-based value chain, from innovation to national and international allocation and delivery.  

Through a series of subsequent comparative analyses aimed at highlighting the viewpoints of public and private stakeholders, these issues are further explored. In addition,  possible solutions are proposed as to foster collaboration between key actors; i.e. public health authorities, governments of industrialized and developing countries, and vaccine industry.

The critical issues are grouped into seven chapters: 1. Accelerating vaccine development; 2. Centralized procurement and alliances; 3. Supply tendering, quality and price; 4. Global accessibility and affordability; 5. Equitable allocation and distribution; 6. Compulsory licensing; 7. A universal pandemic treaty. At the end of each chapter, lessons learned are derived for that topic. The closing chapter summarizes policy and strategy recommendations.”

  1. Manuscript Length

As stated before, we understand that our contribution requires readers to digest a lot of information. Nonetheless, the article is comprehensive and concise but is nevertheless lengthy as it covers a series of key issues that elsewhere have only been partially discussed (and are often biased summarizing the viewpoint of one particular stakeholder). Therefore, this manuscript may be the first publication that brings all key issues and perspectives together so as to effectively handle pandemics and prepare for the next. A holistic approach is indeed necessary to avoid the current mismanagement based on a compartmentalized approach between the upstream processes(i.e. innovation) and downstream processes (i.e. allocation and distribution). Our findings particularly highlight the observed negative induction effects seen between activities, and the resulting sub-optimal way of managing pandemics. It also points to the role of a high-level third party that can be an independent arbiter if given a global mandate.  WHO has accepted the task of ensuring the signing of a future pact between countries.

In general, public policy papers are longer and more text-rich than “hard science” papers. The latter have the advantage of being able to use formulas, and the data can be summarized and displayed in tables and figures. We could have used general flowcharts but decided not to at the risk of being trivial. A quick search regarding the article length reveals that, based on author guidelines, journals with policy interests often require articles to be between 8000-10000 words like ours. We opted for Vaccines/MDPI because of the special Covid issue, without a lower explicit limit.

Finally, we do expect and hope, however, that after this first ‘holistic approach’ contribution there will be many more researchers embarking on exploring any of the critical issues in more depth. This may be the right moment to switch from a qualitative research methodology to a quantitative approach.

  1. Ethical Behaviour

We do not understand your comment, made in the second round, referring to ethical behavior; your quote: Ethics is completely forgotten among "the core activity they are best at". We presume you refer here to the various stakeholders.

If this implies that in our publication we should identify and point fingers at public or private-sector stakeholders by accusing them of being “unethical”, we refute this notion as it is ill-conceived. We distinguish in this matter between potential personal behavior and professional activities carried out in good faith. The latter must be carried out within the law. If any criminal activities should occur then courts will judge.

During our 30 years’ experience in health care policy and management, we have not come across members that behaved intrinsically unethical, or can be labeled as such (which does not exclude controversies and differences in viewpoints). On the contrary, as seen again during the Covid pandemic, there is a great willingness to make a contribution. The crux of the matter is that many work within their own field of competency, and are not aware of the impact their actions may have on any other pandemic-related activities.

In sum, our findings did not discover an intrinsic bad, or perse unfair behavior. We trust that some stakeholder’s lack of knowledge can be remedied by academic publications that shed light on pandemic management issues.